# Goats’ Feeding Supplementation with *Acacia farnesiana* Pods and Their Relationship with Milk Composition: Fatty Acids, Polyphenols, and Antioxidant Activity

**DOI:** 10.3390/ani9080515

**Published:** 2019-08-01

**Authors:** Claudia Delgadillo-Puga, Mario Cuchillo-Hilario, Luis León-Ortiz, Amairani Ramírez-Rodríguez, Andrea Cabiddu, Arturo Navarro-Ocaña, Aurora Magdalena Morales-Romero, Omar Noel Medina-Campos, José Pedraza-Chaverri

**Affiliations:** 1Departamento de Nutrición Animal, Instituto Nacional de Ciencias Médicas y Nutrición Salvador Zubirán (INCMNSZ), CDMX 14080, Mexico; 2Facultad de Química, Universidad Nacional Autónoma de México (UNAM), CDMX 4510, Mexico; 3Dipartimento per la Ricerca nelle Produzioni Animali, Agris, Località Bonassai, 07040 Olmedo Sassari, Italy; 4Departamento de Alimentos y Biotecnología, Facultad de Química, Universidad Nacional Autónoma de México (UNAM), CDMX 04510, Mexico; 5Departamento de Biología, Facultad de Química, Universidad Nacional Autónoma de México (UNAM), CDMX 04510, Mexico

**Keywords:** *Acacia farnesiana* pods, milk, bioactive compounds, radical scavenging, fatty acids

## Abstract

**Simple Summary:**

The study aimed to describe the fatty acids, polyphenols, and antioxidant activity of goat’s milk from five different feeding systems: Grazing; conventional diet; and conventional diet supplemented with 10, 20, and 30 percent of *Acacia farnesiana* pods. Conventional diet showed the highest content of polyunsaturated fatty acids while grazing showed the healthiest fatty acid profile. Similarly, grazing and *A. farnesiana* boosted the polyphenol content. *A. farnesiana* pod meal inclusion in the goats’ diets increased the selective presence of bioactive compounds and the antioxidant activity of goat’s milk while cholesterol content was reduced.

**Abstract:**

Background: Research efforts have focused on the evaluation of the bioactive quality of animal products (milk, cheese, meat, and other by-products) contrasting various feeding strategies coming from different ecological zones. The study aimed to describe the fatty acids (FA), polyphenols (P), bioactive compounds (BC), and antioxidant activity (AA) of goat’s milk. Methods: Dairy goats were fed with five systems: (1) Grazing; (2) conventional diet (CD); (3) CD + 10% of *Acacia farnesiana* (AF) pods; (4) CD + 20% AF; and (5) CD + 30% AF. The fatty acid profile, health promoting and thrombogenic indexes were calculated. Milk extracts were evaluated by HPLC to determent phenolic compounds (gallic, caffeic, chlorogenic, and ferulic acids, catechin, epicatechin, and quercetin). Antioxidant activity of goat’s milk extract was evaluated using 2,2-diphenyl-1-picrylhydrazyl radical (DPPH•), oxygen radical absorbance capacity (ORAC), and the ferric reducing antioxidant power (FRAP) assays. Results: Conventional diet showed the highest content of polyunsaturated fatty acids while grazing showed the best n-6:n-3 and the linoleic:alpha linolenic acid ratio. Similarly, grazing and AF boosted the polyphenol content. Conclusions: *Acacia farnesiana* inclusion in the goats’ diets increased the presence of bioactive compounds and the antioxidant activity while diminishing the cholesterol content of goat’s milk.

## 1. Introduction

In the recent past, research efforts have focused on the evaluation of the bioactive quality of animal products (milk, cheese, meat, and other by-products) contrasting various feeding strategies coming from different ecological zones [1,2,3,4,5,6]. Nevertheless, some of these components are constituents of fat, e.g., polyunsaturated fatty acids (PUFA) with probed beneficial functions for health.

Besides the nutrimental contribution of dairy products, some other health benefits are associated with their consumption, e.g., antioxidant, anti-inflammatory, and analgesic activities [7]. However, the consumption of goat’s milk can generate additional benefits, for example, low allergenicity and better absorption of the lipidic fraction when compared to bovine milk [8]. A lower rate of ruminal fatty acid biohydrogenation is related to a higher amount of beneficial n-3 fatty acids provided by forage base-feeding systems in comparison to cereal-based or total mixed ration of stabulated animals [9]. To counteract this effect, some experiments have been performed to increase the n-3 fatty acid content offering fresh forage-based diets [10] or by implementing grazing feeding systems [3,4,11] to ruminants. Besides, the incorporation of seeds, microalgae [12], fish [13], and seed oil’s [14,15,16,17] supplements in the diet, has positively impacted n-3 fatty acid content of ruminants’ milk.

Some other strategies aimed to decrease the rate of biohydrogenation involving the use of phenolic compounds naturally present in forages. The mechanism to decrease biohydrogenation follows two feasible pathways: The phenolic compounds present in the feedstuffs can be oxidized by the enzyme polyphenol oxidase which can diminish the dietary lipolysis and consequently decrease the biohydrogenation of fatty acids in the rumen liquor [18]. Additionally, the phenolic compounds can generate changes in the population of certain ruminal microorganisms e.g., Butyrovibrio fibrisolvens, further changing ruminal fermentation kinetics [19]. Also, goat production in the arid and semiarid regions currently demands the use of a strategic feeding approach and sustainable use of local feed resources. Nowadays, there is a lack in the amount and quality of feedstuffs for animal productivity [20]. In the central part of Mexico, there are contrasting ecological regions that host plenty and well-adapted plant species traditionally used as animal feed resources. Among those vegetation species, Acacia genus has been documented as an important forage resource for animal feeding and as a rich source of polyphenols [21]. Through in vitro and in vivo experiments, we have demonstrated the effective protective properties of *Acacia farnesiana* (AF) pods against oxidative stress [22]. Plenty of secondary metabolites have been described for AF, e.g., quercetin, gallic acid (GA), catechin, and epicatechin [21,23]. These compounds are known as modulators of different physiological processes, as antioxidants, anti-inflammatory, and anti-bacterial properties [17,21,24]. We hypothesized that AF pods as supplement into the feed of dairy goats, increased the presence of bioactive compounds and antioxidant activity. In this way, we evaluated milk from: (1) Goats under grazing feeding system; (2) goats fed solely conventional diet (CD; concentrate of cereals plus alfalfa hay); (3) goats fed conventional diet supplemented with 10% of AF pods; (4) goats fed conventional diet supplemented with 20% of AF pods; and (5) goats fed conventional diet supplemented with 30% of AF pods.

## 2. Materials and Methods

### 2.1. Experimental Set Up

The experiment was performed in Queretaro, Mexico (20°35′ N, 100°18′ W; 1950 m.a.s.l.), during the summer of 2017. The area is a dry, semiarid climate with isolated rains in winter with an average precipitation of 460 mm. Fifty French Alpine goats (50 ± 5 kg) and a lactation period of 150 days were allocated in to five groups as follows: (1) Grazing goats; (2) conventional diet (CD); (3) CD + 10% of AF pods; (4) CD + 20% of AF pods; and group (5) CD + 30% of AF pods. Animals were housed in herds of ten animals each. The grazing group was allowed to graze and browse freely from 08:00 am to 17:00 pm on 14 ha of rangeland. Grazing animals reared during 8 h/d on 14 ha shrubby rangeland after milking with overnight confinement. The botanical composition of the semiarid rangeland is described in detail by Cuchillo et al. [1]. The main botanical vegetation included forbs (e.g., *Jatropha dioica*, *Celtis pallida*); gramineous species (*Melinis repens*, *Chloris virgata*), leguminous trees (e.g., *A. farnesiana, A. schaffneri, Prosopis laevigata*) and cacti (e.g., *Opuntia affasiacantha*, *O. amyctaea*). Throughout the study, animals fed conventional diet or supplemented conventional diets were kept in full indoor confinement. Feedstuffs for indoor feeding were harvested (lucerne hay) and prepared (grain concentrate) once for the complete experimental period and stored separately. The diets were adjusted to the National Research Council guidelines of energy and protein for angora, dairy, and meat goats in temperate and tropical countries [25]. Rations were elaborated on a daily basis for each indoor animal group (Table 1) and were offered twice per day (in the morning just after milking and at 7:00 am). Water was offered ad libutum in all treatments. All animal groups were milked mechanically once daily at 07:00 am. Milk samples of each animal group were collected separately in seven consecutive days following the adaptation period of 12 days. In total, 10 ± 2 liters for each treatment were collected, labelled, and frozen at −20 °C and lyophilized for further analysis. Protocols for animal housing, animal management, and milk sampling were approved by the Animal Care and Research Advisory Committee (Comité Interno para el Cuidado y Uso de los Animales de Laboratorio (CICUAL) at the INCMNSZ under the registration number NAN-1904-18-19-1.

### 2.2. Acacia farnesiana (AF) Pods

AF pods were harvested in Acatlán de Osorio in the state of Puebla in México; located between 18°04′24” and 18°21′30”, north latitude and 97°55′18” and 98°11′24” west longitude. AF pods were registered with an internal identification number (8757) at the herbarium of the Facultad de Estudios Superiores Cuautitlán at the Universidad Nacional Autónoma de México (UNAM). Manual sampling was performed on aerial parts of AF trees in an area of eight hectares in the lowland Mixtec region of Mexico. Further, pods were completely dried at room temperature and grounded using a knife mill to obtain a particle size of 3–5 mm [21].

### 2.3. Analysis of the Diets

Chemical analysis of the botanical composition of the rangeland was performed according to Cuchillo et al. [1]. Briefly, before the grazing period, samples of the botanical composition were taken in three plots of 20 m by 50 m randomly distributed on the rangeland (14 ha) with ten subplots (0.5 m^2^) per plot, i.e., 30 subplots in total. Both vegetation from rangeland and indoor feeding diets (a mixture of lucerne hay 60% and grain supplement 40%) were ground to a particle size of 1 mm and analyzed as follows: Moisture (oven-drying at 60 °C), fat, crude fiber, and ash content were determined using standard methods [26]. Nitrogen was measured using the micro-Kjeldahl technique [26]. N-free extract was calculated as the difference between 100% and protein (nitrogen factor: 6.25), fat, crude fiber, and ash percentages. Gross energy was determined using the calorimetric Parr bomb (Parr Instrument Company, Moline, IL, USA). All samples were analyzed in triplicate. Cholesterol was measured by the procedure described by Cuchillo et al. [3].

### 2.4. Fatty Acid Profile

It was determined as the recommendation of Cuchillo et al. [3] and Delgadillo et al. [4]. Results were expressed as g/100 of fatty acid methyl esters (FAME). Health promoting index (HPI) was calculated according to Chen et al. [27] while the thrombogenic index (TI) was estimated using the formula from Ulbricht and Southgate [28]. Also, LA:ALA (linoleic acid:alpha linolenic acid); EPA:AA (eicosapentaenoic acid:arachidonic acid) and AA:EPA + DHA (eicosapentaenoic acid:arachidonic acid + docosahexaenoic acid) ratios were calculated.

### 2.5. Goat Milk Extraction

Three subsamples (30 g each) of lyophilized milk were placed into Erlenmeyer flasks (250 mL) with 100 mL of methanol:water (80:20). The flasks were sonicated (Cole-Parmer) during 30 min at room temperature. Further, all extracts were filtered (Whatman No. 4) and washed using 50 mL of the same solvent arrangement. A second extraction was performed to the residue using 100 mL of acetone:water (70:30 v/v) solution. Both filtrates were placed in a flat bottom flask with beaded rim (250 mL) and were concentrated with a vacuum rotary evaporator (IKA-RV 10) at 37 °C and 150 rpm. The concentrate was placed in a test tube and 8 mL of methanol were added. Further, 8 aliquot of 1 mL was allocated into individual Eppendorf tubes for centrifugation (Eppendorf, Ocala, FL, USA; 5804 R; 140× g during 15 min at 4 °C). The supernates were individually stored in amber vials at 4 °C for later analysis.

### 2.6. Total Phenol Content

Total phenolic in milk extracts was determined by the Folin–Ciocalteu colorimetric method described by Chen et al. [29] with some modifications. Briefly, 2 mL of each milk extract were individually placed into a vial (10 mL). Later, 3 mL of hydrochloric acid (HCl; 0.3%) solution was added. Further, an aliquot of 100 µL of the resulted solution was allocated into an amber vial and 2 mL of sodium carbonate (Na_2_CO_3_; 2%) were added. After 2 min, 100 µL of Folin–Ciocalteu’s reagent was added to the mixture. The reaction was left to rest for 30 min at room temperature. Later, 1 mL was placed into quartz cells to be read at a wavelength of 750 nm using a spectrophotometer (Thermo Spectron, Waltham, MA, USA, 60S). The test was done in triplicate for each extract. The concentration was calculated using gallic acid as standard (Sigma-Aldrich, St Luis, MO, USA). Results were expressed as mg gallic acid equivalents (GAE)/100 g of lyophilized milk.

### 2.7. Phenolic Compounds in Goat’s Milk Extract Analyzed by HPLC

Thirty microliters of goat’s milk extract (see above) was analyzed by HPLC to measure the presence of flavonoids and phenolic acids. We used a HPLC 1260 Infinity Agilent Technologies^®^ system, equipped with autosampler 1200 and a quaternary pump of reversed-phase. A Symmetry C18 (4.6 mm × 150 µm; WAT045905, Waters Milford, Milford, MA, USA) column was employed. The conditions of the HPLC were 25 °C, a flow rate of 1 mL/min, and an absorbance of 280 nm. For phenolic acids we used a mobile phase of methanol:phosphoric acid 1M (23:77) and a running time of 40 min. For flavonoids identification we utilized deionized water:acetonitrile:methanol:ethyl acetate:glacial acetic acid (89:6:1:3:1) as mobile phase and running time of 15 min. Gallic acid, caffeic acid, chlorogenic acid, ferulic acid, catechin, epicatechin, and quercetin were used as standards. Each standard (5 mg) was dissolved in 5 mL of methanol. A calibration curve was employed by each standard using five dilutions (1000, 500, 300, 100, and 50 ppm). The software Chem Station Edition 1.06 was employed. Results were expressed as mg 100 g^−1^ of lyophilized milk. All analytical reagents and standards were from Sigma-Aldrich, Steinheim, Germany.

### 2.8. DPPH• Scavenging Activity

For this determination milk extracts were evaluated without dilution. Fifty microliters of 2 mM 2,2-diphenyl-1-picrylhydrazyl (DPPH•) was added to 70 μL of milk extracts and the mixture was vortexed during 2 min. Later, 800 μL of ethanol was added. The solution was let to stand for 2 min at room temperature. Later, it was centrifuged at 1500× g for 2 min. Further, an aliquot of 350 µL was transferred into 96-microwell plates to be read at 517 nm in a Synergy™ HT multimode plate reader (Biotek Instruments, Winooski, VT, USA). The procedure was performed in quadruplicate. Trolox (6-hydroxy-2,5,7,8-tetramethylchroman-2-carboxylic acid; Calbiochem, Billerica, MA, USA) was used as standard to construct the calibration curve (0.05, 0.1, 0.5, and 1.0 mM). Results were expressed as percent (%) of DPPH• scavenged and calculated by the following formula: ((Optical density of control−optical density of compound)/(optical density of control)*100) [30].

### 2.9. Oxygen Radical Absorbance Capacity (ORAC) Assay

The methodology described by Huang et al. [31] was used with some modifications. ORAC assays were performed in a Synergy™ HT multi-mode microplate reader. 2,2′-azobis(2-amidinopropane) dihydrochloride (AAPH), a heat-labile water-soluble azo compound was used as a peroxyl radical generator while catechin (mg/mL) was used as standard. Briefly, 25 μL of milk extract dilution (1:250), 25 μL of 153 mM AAPH solution, and 150 μL of 50 nM fluorescein were placed into black microwell-plates (Costar^®^, Corning, NY, USA). The fluorescence was measured every minute for 90 min at 37 °C using fluorescence filters for an excitation wavelength of 485 nm and an emission wavelength of 520 nm. The ORAC values were calculated employing the software Gen5 Version 2.01 (Bio Tek Instruments Inc., Winooski, VT, USA).

### 2.10. Ferric Reducing Antioxidant Power (FRAP) Test

For this determination, the methodology of Benzie and Strain [32] was followed. Two sets of extracts were prepared. For the first set, the extracts were analyzed without any dilution while for the second set, a dilution of 1:10 was employed. Briefly, 30 μL of milk extract or milk extract solution plus 300 μL of FRAP solution (1.66 mM FeCl_3_, 0.83 mM 2,4,6-Tris(2-pyridyl)-s-triazine (TPTZ), in 300 mM acetate buffer (pH 3.6)) were placed into 1.5 mL microcentrifuge tubes. After 15 min of incubation at room temperature the tubes were centrifuged for 2 min at 1500× g and 300 µL of supernatant were placed into microwell-plates (Costar^®^ Transwell^®^) to read the absorbance at 593 nm. A calibration curve of FeSO_4_ (0.025, 0.05, 0.1, 0.25, 0.5, 0.75, and 1 mM) was constructed to estimate the ferric reducing antioxidant power of the samples. The results were expressed as millimoles of FeSO_4_/L of milk.

### 2.11. Statistical Analysis

Results of measurements per sample were averaged before further statistical analysis. The statistical model used was an analysis of variance for the comparison of independent groups. Because the data were not assumed to be normal and the sample sizes were small, non-parametric statistics were used. The Kruskal–Wallis test was used to establish differences among treatments. Comparison of the medians with a significant difference was set to *p* < 0.05. Mann–Whitney U test signed ranks test for related pairs of portions was used to identify such differences in IBM SPSS Statistics 23.0 program (Armonk, NY, USA). Data are present as the mean ± standard deviation (SD). Pearson’s correlation was constructed among the antioxidant activity and the bioactive compounds concentration on the samples (*p* < 0.05) using the GraphPad Prism 7.0a (San Diego, CA, USA).

## 3. Results

### 3.1. Analysis of the Diets and Milk

We observed that crude protein (9.22 g/100 g) and ether extract (1.68 g/100 g) contents of feeding in the grazing system were slightly lower than the rest of the diets. In contrast, carbohydrate content showed the top value (56 g/100 g) in the grazing diet (Table 1).

The goat’s milk composition did not show statistical differences for water content, protein, fat, carbohydrates, ash, and energy. However, this observation was the opposite for cholesterol, where statistical difference was detected among treatments (Table 2).

### 3.2. Fatty Acid Methyl Esters (FAME) and Health and Risk Indices

The fatty acids stearic (C18:0), linoleic (C18:2; *cis*-9, *cis*-12), linolelaidic (C18:2; *trans*-9, *cis*-12), and alpha linolenic (C18:3 n-3) increased (*p* < 0.001) in milk when goats were fed with the conventional diet. However, the largest concentration of conjugated linoleic acid (C18:2 *cis*-9, *trans*-11 fatty acid-CLA) was found in milk from grazing goats followed by milk from goats fed conventional diets supplemented with 20% and 30% AF. Grazing (69.7 g/100 g FAME) and CD + 30% AF pods (69.4 g/100 g FAME) increased the content of saturated fatty acids (SFA) of goat’s milk. In contrast, milk from the grazing system showed the lowest n-6 fatty acid content (2.9 g/100 g). In the same line, grazing decreased the monounsaturated fatty acids (MUFA) content (26.2 g/100 g) in relation to conventional diet, CD + 10% and CD + 20% AF pods. On the other hand, conventional diet supplemented with 30% AF increased the SFA content (69.4 g/100 g) while decreased the MUFA content (25.9 g/100 g) and showed the highest TI (3.6) among all treatments (Table 3). Conventional diet resulted with the highest content of PUFA (5.6 g/100 g), n-6 fatty acids (4.5 g/100 g), and n-3 fatty acids (0.96 g/100 g). Grazing system showed the lowest (best) n-6: n-3 ratio (3.3), while the conventional diet supplemented with 20% and 30% AF averaged the top (worst) values (5.1 and 5.7). Linoleic (LA; C18.2 n-6)/alpha-linolenic acids (C18:3 n-3) ratio increased with grazing feeding followed by conventional diet (5.07); CD + 10% (6.05), CD + 20% (5.97), CD + 30% (6.99) AF pods. The best EPA (C20:5 n-3)/AA (C20:4 n-6) ratio (0.32) was found in the conventional diet in comparison to the rest of the feeding systems (*p* < 0.05).

### 3.3. Total Phenolic Content and Bioactive Compounds

The inclusion of AF pods in the goats’ diets increased the total phenolic content in the milk (Figure 1). The highest concentration of phenols was found in the CD + 30% treatment (305.5 mg of GAE/L of milk) while CD was the lowest in this respect (159.4 mg of GAE/L of milk). Grazing system also increased this parameter but to a lesser extent than the CD + 30% treatment; however, showed higher values that CD + 10% treatment (179.7 mg of GAE/L of milk) and CD + 20% treatment (200.3 mg of GAE/L of milk).

The analysis of the bioactive compounds was divided into phenolic acids and flavonoids. For both chemical groups (from Figure 2A–D) gallic, chlorogenic, ferulic acids, and catechin were detected in all treatments except in conventional diet. AF pods inclusion tended to elevate the concentration of phenolic acids and catechin. However, the maximum concentration of those compounds was recorded by the grazing system (1.28, 12.3, 5.61, and 4.21 mg/100 mL of milk, respectively).

### 3.4. Antioxidant Activity

As the concentration of bioactive compounds in the milk extracts increased, the DPPH• scavenging capacity tended to increase with the supplementation of AF pods. Moreover, the milk extracts derived from animals that grazed have the highest performance (*p* < 0.05) to scavenge these free radicals (Figure 3A). A positive correlation was found between the DPPH• antioxidant activity and the bioactive compounds concentrations (Figure 4A1–A4). The same pattern was observed for ORAC (Figure 4B1–B4) and FRAP (Figure 4C1–C4) tests, with some differences. For ORAC, grazing system and CD + 30% AF pods boosted the antioxidant activity while lower levels of AF pods inclusion (10% and 20%) and conventional diets, did not showed this desirable effect in the milk (Figure 3B). Similarly, data from FRAP assay imitated the outcomes of DPPH• and ORAC tests. Here, CD + 30% AF pods showed the best performance (92.4 µM of FeSO_4_/100 mL of milk), followed by the grazing (61.8 µM of FeSO_4_/100 mL of milk), CD + 20% AF pods (56.6 µM of FeSO_4_/100 mL of milk), conventional diet (47.9 µM of FeSO4/100 mL of milk), and CD + 10% AF pods (47.4 µM of FeSO_4_/100 mL of milk) (Figure 3C).

## 4. Discussion

In the present study, the increasing supplementation of AF pods meal tended to diminish the protein and energy content of the diets but the other parameters did not show any substantial change. Some detrimental effects might be observed on the overall milk yield and the body condition of goats caused as side-effects due to the inclusion of AF pods and the lower proteic and energetic content of the diets. However, the values of both parameters are close to the requirements recommended for lactating dairy goats [32]. Though this constrains, the utilization of local resources as AF for animal feeding promotes sustainable practices which enables a more resilient way to produce animal food deliveries. Therefore, longer set up studies should be conducted to examine the effects on AF pods to elucidate the effects of their inclusion in the goat’s milk yield along the curve of lactation and in the body condition score.

The lack of differences for water content, protein, fat, carbohydrates, ash, and energy parameters among treatments in goats’ milk could be due to the physiological compensation of the animals and the mammary gland which are directed to deplete stored nutrients to keep milk composition and production. This effect might impact body condition if supplementation is not properly amended or a long deprivation of nutrients occurs. Therefore, studies with longer extent should be tested to corroborate this assumption. Cholesterol in CD was the top value while the inclusion of AF and grazing treatment tended to diminished this value (Table 2). In this regard, a first analysis of AF pods was performed by our research group, showed that alcoholic extracts of AF pods contained several phytosterol molecules namely, campesterol, stigmasterol, and gamma-sitosterol [24]. A later study of our research group was focused in the composition of AF pods through nuclear magnetic resonance, showed an important number of stigmasterol derived compounds such as 22E-stigmasta-5, 22-dien-3β-ol, 22E-stigmasta-5,22-dien-3β-acetyl, stigmasta-5,22-dien-3β-O-D-glucopyranoside, and stigmasta-5,22-dien-3β-O-D-tetraacetylglucopyranoside [33]. Phytosterols appear to displace cholesterol from micelles and reduce the absorption of cholesterol in the intestine. However, some phytosterols with Δ22-unsaturation, like stigmasterol, broadly found in AF, can competitively inhibit sterol Δ24-reductase in two human cell lines (epithelial Caco-2 and promyelocytic leukemia—HL-60) in a dose-dependent manner. Sterol Δ24-reductase is an important enzyme in cholesterol synthesis, it catalyzes the conversion of lanosterol (considered the first steroid in the pathway), into desmosterol or 7-dehydrocholesterol, both products which are later turned into cholesterol [34]. Inhibition of this enzyme (sterol Δ24-reductase) through stigmasterol from AF pod supplementation might reduce the de novo cholesterol production in mammary gland of goats, ultimately lowering the cholesterol content in milk. This helps to explain the differences found in cholesterol milk among treatments. 

A large body of literature has proved that ruminants fed under rangeland conditions where the fatty acid profile is shifted to have larger shares of MUFA and PUFA and lower SFA in comparison to conventional feeding [3,4,11]. However, the results of the present study contrast this trend. SFA content in the grazing system and the group fed CD + 30% AF pods were the two treatments with the highest value. On the other hand, conventional diet, CD + 10% AF and CD + 20% AF are comparable in this parameter value (SFA). Opposite to the values of SFA, MUFA were found to have lower values in the grazing system and with the inclusion of CD + 30% AF pods while the highest values were for conventional diet, CD + 10% AF, and CD + 20% AF groupings. The good quality of the forages included in the diets offered in the install diets (especially alfalfa), could be responsible for these observations, since the hay of alfalfa is a source of MUFA and PUFA. Also, although AF pods are not a rich source of lipids (around of 1.5 g of lipids/100 g dry matter), the composition of milk was affected by the inclusion on their meal in the goats’ diets. Likely, this effect is linked to the modifications in the ruminal kinetics as a result of the rich bioactive compounds found in these seeds. In vitro studies have proved that vegetation with high phenolic content directly affects the biohydrogenation and lipolysis metabolism of fatty acids. Greater phenolic content is associated with lower presence of LA and higher ALA syntheses [18]. As a consequence of this phenomenon, the conventional diet showed the highest PUFA content and it was different from the rest of the groups. The selective ALA synthesis is desirable, as this fatty acid is the precursor of n-3 fatty acids (DHA and EPA) with high positive impact on human health to prevent the incidence of chronic maladies. The result associated with this effect, is the LA/ALA ratio, where the grazing treatment developed the best score (3.62; the lowest the best) among all the groups evaluated. This observation is also closely related to the stearic acid (C18:0) content (smallest) in the grazing system, which is the product of the biohydrogenation of ALA and LA during the lipid ruminal metabolism. Such metabolites are recently associated with healthy activities tested in animal models with potential to be translated to human wellbeing. The EPA/AA and n-6:n-3 ratio indicated that grazing would have the best responses to support benefits for human health. However, the highest n-3 fatty acids content and the HPI was registered in the conventional diet. In marked contrast to conventional diet, the best thrombogenic index was found in comparable value in milk from CD, CD + 10% AF, and CD + 20% AF diverged to the less favorable values of milk from the grazing system and CD + 30% AF groups. With respect to CLA, the grazing system recorded the highest value. Dissimilar shares of fatty acids are the result of the varied metabolism and kinetics of rumen as well as the profile dietetic of fatty acids [18]. Wherefore, improving the utilization of local vegetation (e.g., *A. farnesiana*) for livestock production would lead to increase the functional quality of animal products (milk, meat, and by-products), supporting a healthy and appropriate diet for humans while promoting preventive actions aimed to reduce chronic disorders.

Regarding the antioxidant activity of milk extracts analyzed by DPPH•, ORAC, and FRAP assays, additive effects were observed when goats consumed increasing proportions of AF pods. It is important to mention that the concentration of total phenolic content from the grazing system correlated with the antioxidant activity evaluated by these antioxidant tests. A possible explanation of this outcome is the high content of bioactive polyphenolic constituents of AF as well as to its high antioxidant activity [5,21]. This additive effect was observed in goat’s milk from CD + 30% AF was tested in FRAP and ORAC assays. However, in DPPH• assay this trend was not observed. This effect was also observed by Bhoyar et al. [35]. The distinct responses observed among the antioxidant assays may be explained by the fact that these methods have differences in their reaction mechanisms, conditions, and even in detection parameters. The antioxidant capacity measurements relay on methods based on electron transfer (ET) reaction such as DPPH, FRAP, and Trolox equivalent antioxidant capacity (TEAC) assays, or hydrogen atom transfer (HAT) reaction such as ORAC, crocin bleaching, total peroxyl radical trapping antioxidant parameter (TRAP), and luminol-chemiluminescence based peroxyl radical scavenging capacity (LPSC) assays [36]. DPPH• assay indicates the ability of samples to reduce the radicals by measuring the decrease in the absorbance while the FRAP test is based on the capacity of compounds to reduce Fe3 + to Fe2 + which forms a colored complex with TPTZ at pH 3.6 and ORAC assay measures the capacity of molecules to trap peroxyl radicals and thus delays the peroxyl radical-dependent decrease of fluorescence intensity of fluorescein at pH 7.4. Those differences in the assays may explain the differing results of the same evaluated extracts. Additionally, the intrinsic characteristics of the samples as complexity and solubility of its components may contribute to these differences [30]. The analyses of samples employing different antioxidants assays may show similar or different results. Data obtained from antioxidant analysis by FRAP (ET reaction) and ORAC (HAT reaction) methods show a similar response for different antioxidants from contrasting origin. Moreover, [37] observed a correlation between ORAC and FRAP assays but there was no correlation among ORAC and TEAC, although FRAP and TEAC methods are based on ET reaction.

## 5. Conclusions

No clear effects of feeding treatment were observed to modify the fatty acid profile of goat’s milk. High n-3 fatty acid concentration, a reasonable antioxidant activity as well as the presence of bioactive compounds were observed in milk from grazing/browsing goats. Also, PUFA concentration of goat’s milk was increased by the good quality of feedstuffs utilized in the conventional diet. On the other hand, *Acacia farnesiana* pods inclusion in the goats’ diets increased the selective presence of bioactive compounds and the antioxidant activity of milk. *Acacia farnesiana* supplementation diminished the cholesterol content in milk.

## Figures and Tables

**Figure 1 animals-09-00515-f001:**
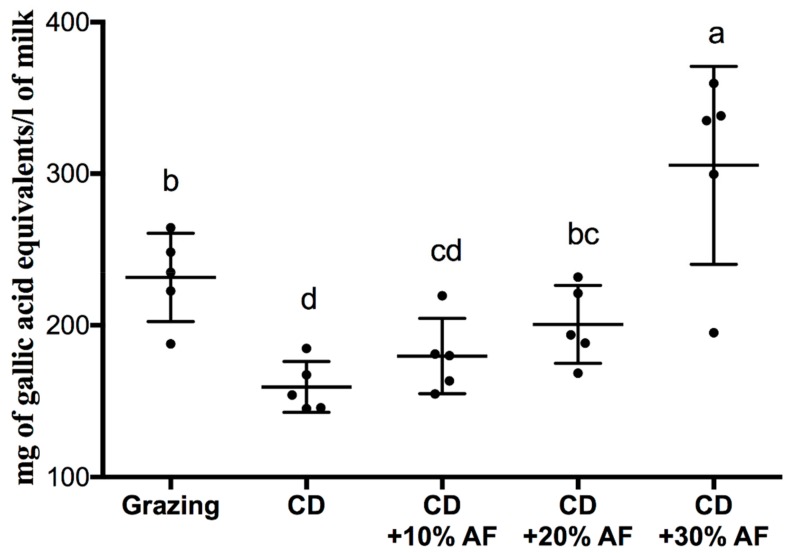
Total polyphenols of freeze-dried milk from goats fed conventional diet versus diets supplemented with graded levels of *Acacia farnesiana* pods (AF). CD = Conventional diet. ^a,b,c,d^ Means with different letters indicate differences (*p* < 0.05) among treatments.

**Figure 2 animals-09-00515-f002:**
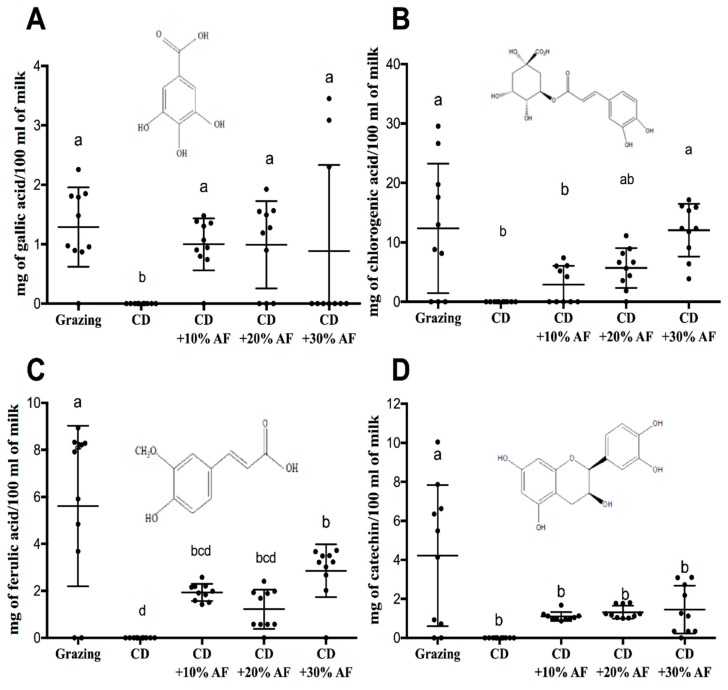
Bioactive compounds (**A**–**D**) in milk from goats fed conventional diet versus diets supplemented with graded levels of *Acacia farnesiana* pods (AF) by HPLC. CD = Conventional diet. ^a,b,c,d^ Means with different letters indicate differences (*p* < 0.05) among treatments.

**Figure 3 animals-09-00515-f003:**
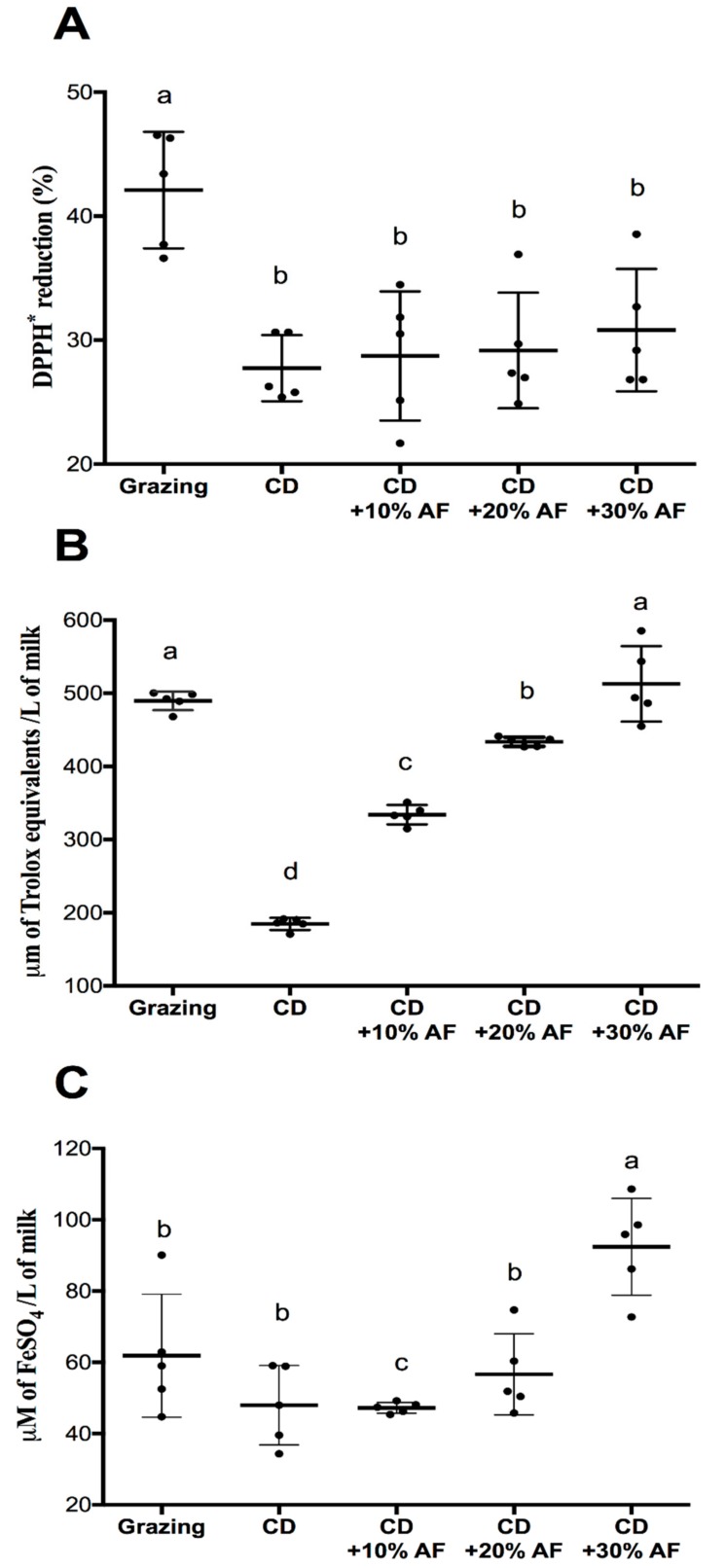
Total antioxidant activity of milk extracts from goat’s fed under grazing system, conventional diet (CD), or CD supplemented with different levels of *Acacia farnesiana* pods (AF). DPPH• scavenging capacity (**A**). Oxygen radical absorbance capacity (ORAC) assay (**B**) and ferric-reducing antioxidant power (FRAP) assay (**C**). ^a,b,c,d^ Means with different letters indicate differences (*p* < 0.05) among treatments.

**Figure 4 animals-09-00515-f004:**
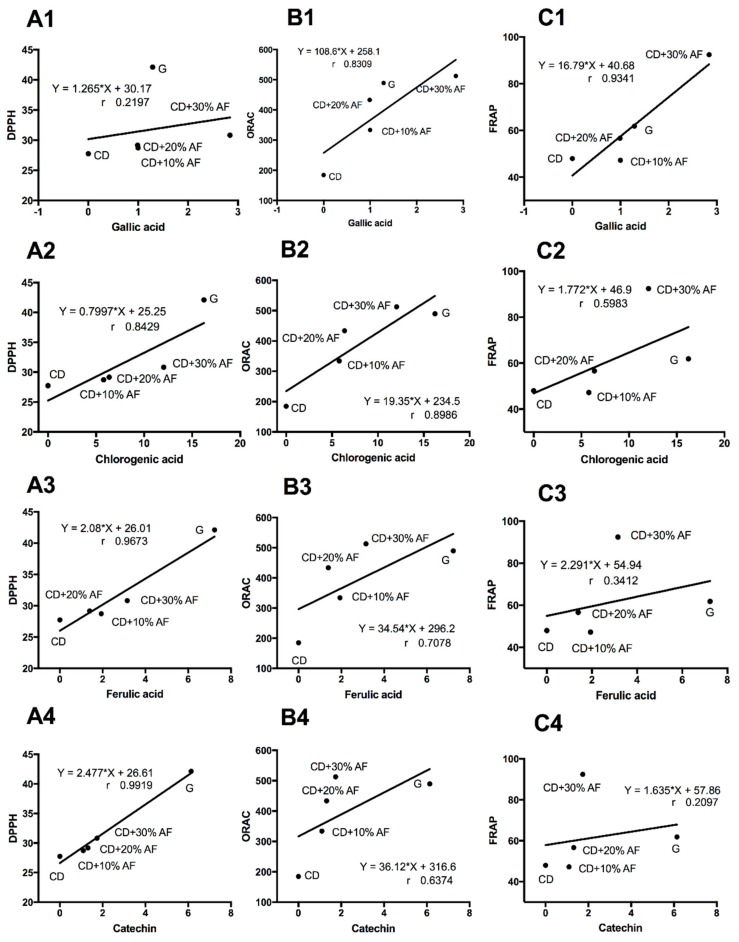
Pearson correlations among antioxidant activity and bioactive compounds concentrations from goat’s milk fed under grazing system (G), conventional diet (CD), or CD supplemented with different levels (10%, 20%, and 30%) of *Acacia farnesiana* pods (AF). DPPH• scavenging capacity assay (**A1**–**A4**). Oxygen radical absorbance capacity (ORAC) assay (**B1**–**B4**) and ferric-reducing antioxidant power (FRAP) assay (**C1**–**C4**).

**Table 1 animals-09-00515-t001:** Goats’ diets ingredients, experimental diets (g/100 g) from semiarid systems in Mexico.

Goats’ Diets Ingredients	Grazing ^1^	Conventional Diet (CD)	CD + 10% AF	CD + 20% AF	CD + 30% AF
	Percentage of inclusion				
Lucerne hay		60	54	48	42
Grain supplement ^2^	-	40	36	32	28
*Acacia farnesiana* pods meal	-	-	10	20	30
Chemical composition of experimental diets (g/100 g of dry matter basis)					
Dry matter	95.5	98.02	97.69	98.17	96.35
Crude protein (N*6.25)	9.22	15.00	15.38	14.98	13.85
Ether extract	1.68	3.42	3.08	2.57	3.40
Carbohydrates	55.88	45.13	45.62	49.53	51.76
Ash	11.11	15.34	14.85	13.14	10.21
Fiber crude	22.01	19.13	18.76	17.95	17.13
Gross energy (kcal/g)	3.24	4.67	4.72	4.64	3.77

AF = *Acacia farnesiana* pods. ^1^ Obtained from mean values of forbs, leguminous trees, and cactaceous vegetation encountered in the grazing/browsing areas. ^2^ Grain concentrate = rolled corn 55%; wheat bran 17%; barley 15%; soybean 9.3%; vitamins and minerals 3.7%. Mixture of vitamins and minerals content in grams per kilogram: vit D3 4000 UI, vit A 32 000 UI, vit E 100 g, vit B12 40 g, vit K 34 g, vit B1 8.0 g, vit B2 8.0 g, vit B6 8.0 g, zinc 60 g, manganese 43 g, panthotenic acid 40 g, copper 6 g, iron 4 g, cobalt 1 g, biotin 200 mg, iodine 32 mg, and selenium 8 mg.

**Table 2 animals-09-00515-t002:** Goat’s milk chemical composition (g/100 g) from semiarid systems in Mexico. AF = *Acacia farnesiana* pods.

Parameters	Grazing	Conventional Diet (CD)	CD + 10% AF	CD + 20% AF	CD + 30% AF
Water content	88.39	88.10	88.42	88.63	89.09
Protein (N*6.38)	3.59	3.61	3.42	3.34	3.50
Fat	3.36	3.98	3.55	3.25	2.84
Carbohydrates	3.88	4.55	3.82	4.10	3.72
Ash	0.78	0.75	0.79	0.69	0.85
Gross energy (kcal 100/g)	0.60	0.64	0.61	0.59	0.54
Cholesterol (mg/100 g)	14.80 c	18.10 a	15.21 b	13.33 d	11.65 e

^a,b,c,d,e^ Means with different letters indicate differences (*p* < 0.05) among treatments.

**Table 3 animals-09-00515-t003:** Fatty acids profile (g/100 g fatty acid methyl esters, FAME) and health/risk indices in the goat’s milk system from semiarid systems in Mexico.

Fatty Acids	Grazing	Conventional Diet (CD)	CD + 10% AF	CD + 20% AF	CD + 30% AF
C8:0 Caprylic	0.11 ± 0.2 ab	0.23 ± 0.07 a	ND	0.14 ± 0.02 b	0.07 ± 0.01 c
C10:0 Capric	2.93 ± 0.04 b	3.65 ± 0.14 a	2.50 ± 0.09 c	1.67 ± 0.08 d	1.94 ± 0.11 e
C11:0 Undecanoic	0.05 ± 0.01 a	0.04 ± 0.01 b	0.03 ± 0.01 b	0.04 ± 0.04 b	0.03 ± 0.01 b
C12:0 Lauric	3.51 ± 0.04 b	2.80 ± 0.13 d	3.68 ± 0.06 a	3.65 ± 0.09 a	3.28 ± 0.15 c
C13:0 Tridecanoic	0.08 ± 0.1 ab	0.07 ± 0.01 a	0.09 ± 0.01 b	0.08 ± 0.1 ab	0.08 ± 0.1 ab
C14:0 Myristic	12.64 ± 0.2 b	12.13 ± 0.3 c	11.80 ± 0.2 d	12.74 ± 0.3 b	13.97 ± 0.1 a
C15:0 Pentadecanoic	1.12 ± 0.02 b	0.98 ± 0.03 d	1.29 ± 0.02 a	1.28 ± 0.02 a	1.03 ± 0.03 c
C15:1 *cis*-10 Pentadecanoic	0.35 ± 0.03 b	0.30 ± 0.01 c	0.42 ± 0.01 a	0.41 ± 0.02 a	0.33 ± 0.02 b
C16:0 Palmitic	38.26 ± 0.5 b	29.94 ± 0.6 d	31.91 ± 0.9 c	32.09 ± 0.5 c	41.36 ± 0.5 a
C16:1 *cis*-9 hexadecanoic	0.78 ± 0.05 c	0.89 ± 0.01 b	0.82 ± 0.06 c	0.90 ± 0.04 b	1.00 ± 0.03 a
C17:0 Heptadecanoic	0.83 ± 0.03 c	0.91 ± 0.03 b	1.02 ± 0.07 a	0.75 ± 0.7 cd	0.71 ± 0.02 d
C17:1 *cis*-10-Heptadecanoic	0.37 ± 0.02 d	0.41 ± 0.01 c	0.50 ± 0.5 a	0.44 ± 0.02 b	0.35 ± 0.01 e
C18:0 Stearic	9.68 ± 0.54 b	11.75 ± 0.09 a	9.89 ± 0.44 b	9.77 ± 0.09 b	6.57 ± 0.21 c
C18:1 *cis*-9 Oleic	24.65 ± 0.1 b	29.76 ± 0.3 a	30.58 ± 0.9 a	30.42 ± 0.6 a	24.12 ± 0.1 c
C18:2 n-6 *cis*-9, *cis*-12 Linoleic (LA)	2.42 ± 0.06 d	3.97 ± 0.02 a	3.22 ± 0.09 c	3.33 ± 0.11 cb	3.36 ± 0.03 b
C18:2 *trans*-9, *cis*-12 Linolelaidic	0.20 ± 0.02 b	0.25 ± 0.02 a	0.20 ± 0.01 b	0.18 ± 0.01 b	0.13 ± 0.04 e
C18:2 (CLA)	0.29 ± 0.03 a	0.20 ± 0.02 b	0.22 ± 0.03 c	0.23 ± 0.01 c	0.23 ± 0.01 c
C18:3 n-3 Alpha linolenic (ALA)	0.67 ± 0.04 b	0.78 ± 0.04 a	0.53 ± 0.03 c	0.57 ± 0.09 bc	0.48 ± 0.02 d
C18:3 n-6 Gamma linolenic	0.02 ± 0.05 c	0.04 ± 0.01 b	0.04 ± 0.01 b	0.07 ± 0.01 a	0.03 ± 0.01 b
C20:0 Arachidic	0.32 ± 0.01 b	0.25 ± 0.02 c	0.35 ± 0.02 a	0.38 ± 0.03 a	0.26 ± 0.05 c
C20:1 *cis*-11-eicosanoic	0.07 ± 0.01 a	0.07 ± 0.01 a	0.07 ± 0.01 a	0.05 ± 0.04 b	0.05 ± 0.01 b
C20:2 n-6	ND	0.03 ± 0.04 a	0.05 ± 0.02 a	0.04 ± 0.04 a	ND
C20:3 n-6 *cis*-8,11,14-eicosatrienoic	0.02 ± 0.005	0.02 ± 0.001	0.02 ± 0.003	0.02 ± 0.003	ND
C20:3 n-3 *cis*-11,14,17-eicosatrienoic	0.08 ± 0.02 a	0.07 ± 0.0 a	0.09 ± 0.01 a	0.09 ± 0.01 a	0.05 ± 0.01 b
C20:4 n-6 Arachidonic (AA)	0.24 ± 0.01 b	0.20 ± 0.02 c	0.32 ± 0.03 a	0.29 ± 0.02 a	0.31 ± 0.01 a
C20:5 n-3 Eicosapentaenoic (EPA)	0.06 ± 0.1 ab	0.06 ± 0.01 ab	0.07 ± 0.01 a	0.05 ± 0.04 b	0.06 ± 0.1 ab
C22:6 n-3 Docosahexaenoic (DHA)	0.06 ± 0.01 a	0.04 ± 0.003 b	0.07 ± 0.01 a	0.07 ± 0.01 a	0.08 ± 0.01 a
C23:0 Tricosanoic	0.05 ± 0.04 b	0.05 ± 0.01 b	0.06 ± 0.01 a	0.05 ± 0.03 b	0.04 ± 0.01 c
C24:0 Lignoceric	0.04 ± 0.03 b	0.03 ± 0.04 c	0.05 ± 0.05 a	0.06 ± 0.01 a	ND
LA/ALA	3.62 ± 0.24 d	5.07 ± 0.26 c	6.05 ± 0.37 b	5.97 ± 0.90 bc	6.99 ± 0.2 ab
EPA/AA	0.24 ± 0.04 b	0.32 ± 0.03 a	0.22 ± 0.01 b	0.18 ± 0.02 b	0.20 ± 0.02 b
DHA/AA	0.26 ± 0.05	0.20 ± 0.01	0.23 ± 0.03	0.26 ± 0.01	0.25 ± 0.02
AA/EPA + DHA	2.07 ± 0.38	1.94 ± 0.14	2.23 ± 0.13	2.29 ± 0.16	2.21 ± 0.19
ΣSFA ^1^	69.73 ± 0.2 a	62.74 ± 0.3 b	62.80 ± 1.0 b	62.83 ± 0.8 b	69.42 ± 0.2 a
ΣMUFA ^2^	26.22 ± 0.1 b	31.62 ± 0.3 a	32.39 ± 1.0 a	32.23 ± 0.6 a	25.85 ± 0.1 c
ΣPUFA ^3^	4.05 ± 0.08 c	5.64 ± 0.08 a	4.81 ± 0.05 b	4.95 ± 0.22 b	4.74 ± 0.07 b
Σn-6	2.89 ± 0.05 c	4.48 ± 0.07 a	3.81 ± 0.06 b	3.93 ± 0.13 b	3.83 ± 0.04 b
Σn-3	0.87 ± 0.07 b	0.96 ± 0.10 a	0.77 ± 0.04 c	0.79 ± 0.10 bc	0.67 ± 0.03 d
n-6/n-3	3.34 ± 0.27 c	4.70 ± 0.27 b	4.99 ± 0.28 b	5.05 ± 0.5 ab	5.70 ± 0.21 a
HPI ^4^	0.32 ± 0.01 c	0.46 ± 0.01 a	0.45 ± 0.1 ab	0.43 ± 0.02 b	0.30 ± 0.03 d
TI ^5^	3.47 ± 0.06 b	2.55 ± 0.04 c	2.61 ± 0.01 c	2.65 ± 0.12 c	3.63 ± 0.05 a

^a,b,c,d,e^ Data are expressed as means (n = 3) ± SE. Columns with different letters indicate significant differences, *p* < 0.05. ^1^ ΣSFA = C8:0, C10:0, C11:0, C12:0, C13:0, C14:0, C15:0, C16:0, C17:0, C18:0, C20:0, C23:0, C24:0. ^2^ ΣMUFA = C15:1, C16:1, C17:1, C18:1, C20:1. ^3^ ΣPUFA = C18:2, C18:2, C18:2 (CLA), C18:3n-3, C18:3n-6, C20:2n-6, C20:3n-6, C20:3n-3, C20:4n-6, C20:5n3, C22:6n-3. ^4^ HPI = (n-6PUFA + n-3PUFA + MUFA)/((C12:0 + (4 × C14:0) + C16:0)). ^5^ TI = (C14:0 + C16:0 + C18:0)/[(0.5 MUFA) + (0.5n-6PUFA) + (3n-3PUFA) + (n-3PUFA/n-6PUFA)]. AF = *Acacia farnesiana* pods. ND = not detected. CLA = Conjugated linoleic acid isomers (*cis*-9, *trans-*11; *trans-*9, *cis-*11; *trans-*10, *cis-*12, *cis-*10, *cis-*12).

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
