# Peer review of "Goats’ Feeding Supplementation with Acacia farnesiana Pods and Their Relationship with Milk Composition: Fatty Acids, Polyphenols, and Antioxidant Activity"

_animals, 2019, doi:10.3390/ani9080515_

Round 1

Reviewer 1 Report

Additional remarks:

No. of line: 

50. Newest marine algae report should be added!

71. healthier

77. What feeding standards were used for diet calculation? NRC? Please specify (not only in line 297!)

92. vitamin and minerals?

83-93. time duration: 0700; 0800-1700 8h duration? after milking?

Table 1: In grazing treatment, the dry matter value is very high! Why?

If possible, data about fatty acid composition of the experimental diets (forages and pasture) should be included in Table 1.

88. How the animals were housed? Individually penned or in the herd? Please specify

How offer the forages (incl. AF and grains) to the animals? 1x or 2x per day?

119. g/100g

Table 2: ether exract – fat content? Carbohydrates – lactose? crude protein – protein

Table 2: “… milk composition not show great differences…” but in the Table found differences, such as milk fat content: CD: 3.98 vs. CD+30%AF: 2.84! Moreover, compare to 313. line!

218. Rumenic acid should be used belongs to conjugated linoleic acid!

232. EPA/AA?

Table 3: title of table: arid or semiarid region in Mexico?!

Figure 2: Very huge the SD values of all bioactive compounds in grazing treatment (and in case of gallic acid in SD+30%). Why?

294-295. Condition scoring and milk yield: no information in material methods and results sections! Please, give details if they were available or modify this sentence!

331. rumenic acid belongs to CLA.

331. Contents of rumenic acid reported in Table 3 are very low in all treatments. How could be justified this result?

Author Response

Referee: 1

Comments to the Author

50. Newest marine algae report should be added!

Answer: A new quotation was included.

71. healthier

Answer: The whole sentence was rephrased, now the word “healthy” was used instead.

77. What feeding standards were used for diet calculation? NRC? Please specify (not only in line 297!)

Answer: The NRC Nutrient Requirement of Goats: Angora, dairy, and meat goats in temperate and tropical countries guidelines were included. The complete description is in the material and methods section.

92. vitamin and minerals?

Answer: To give more clarity, vitamins and minerals were described in detail in the foot of the Table 1.

83-93. time duration: 0700; 0800-1700 8h duration? after milking?

Answer: in the paragraph it is stated that the animals were allowed to graze during eight hours per day. Unfortunately, a complete ethogram was not carried out to assure that the complete time was used for such activity. Therefore, is only mention that were “allowed” to graze. “After milking” is now included in the paragraph.

Table 1: In grazing treatment, the dry matter value is very high! Why?

Answer: the humidity value reported is the analytical humidity. i.e. This humidity is in dry matter basis, therefore this values is high. This is now specified in the manuscript.

If possible, data about the fatty acid composition of the experimental diets (forages and pasture) should be included in Table 1.

Answer: We do not have the possibility to perform an additional determination in diets. At the moment we have not the methodology to perform and validate this procedure on plant tissues. 

88. How the animals were housed? Individually penned or in the herd? Please specify. How offer the forages (incl. AF and grains) to the animals? 1x or 2x per day?

Answer: This section was improved by describing the details of the housing and the meal times per day.

119. g/100g

Answer: in this point specifically, we could not follow the suggestion of the reviewer.

Table 2: ether exract – fat content? Carbohydrates – lactose? crude protein–protein

Answer: the table was modifying and the concepts were changed accordingly.

Table 2: “… milk composition not show great differences…” but in the Table found differences, such as milk fat content: CD: 3.98 vs. CD+30%AF: 2.84! Moreover, compare to 313. line!

Answer: the paragraph was rephrased and a section in conclusion was added accordingly.

218. Rumenic acid should be used belongs to conjugated linoleic acid!

Answer: in our methodology we used a mixture of CLA isomers as standards, therefore we report this value of CLA as a whole. It was modified properly to avoid misunderstandings. CLA= conjugated linoleic acid isomers (cis-9, trans-11; trans-9, cis-11; trans-10, cis-12, cis-10, cis-12).

232. EPA/AA?

Answer: this ratio is correct. It has been previously reported to better characterize the fatty acid profile. (Abe, S., Sugimura, H., Watanabe, S., Murakami, Y., Ebisawa, K., Ioka, T., … Inoue, T. (2018). Eicosapantaenoic acid treatment based on the EPA/AA ratio in patients with coronary artery disease: follow-up data from the Tochigi Ryomo EPA/AA Trial in Coronary Artery Disease (TREAT-CAD) study. Hypertension Research, 41(11), 939–946. http://doi.org/10.1038/s41440-018-0102-9)

Table 3: title of table: arid or semiarid region in Mexico?!

Answer: this sentence is certainly incorrect; the correct word is semiarid. It was modified as suggested.

Figure 2: Very huge the SD values of all bioactive compounds in grazing treatment (and in case of gallic acid in SD+30%). Why?

Answer: in the cases where the value was not detected, a “0” value was allocated. Therefore, we observed such differences within a single treatment.

294-295. Condition scoring and milk yield: no information in material methods and results sections! Please, give details if they were available or modify this sentence!

Answer: we did not measure body score and milk yield, this is solely a suggestion to continuing later studies on those topics in relation to AF supplementation.

331. rumenic acid belongs to CLA.

Answer: this observation is certainly correct. It was modified as we explain above.

331. Contents of rumenic acid reported in Table 3 are very low in all treatments. How could be justified this result?

Answer: rumenic acid which is part of our total value for CLA, is reported in table 3. This fatty acid is part of the total PUFA content; then low PUFA values combined with high MUFA and SFA content, consequently yielded low shares of CLA. This is the reason that helps to explain the reduced percentage in all treatment for this fatty acid.

Reviewer 2 Report

A very interesting scientific article worth publishing.
Critical remarks:
• Give how the chemical composition of milk was analyzed (Table 2)
• Why was there no fat concentration in the milk analyzed (Table 2)?
• How was the cholesterol content in milk determined?
• L211-L212: The sentence “The goats' milk composition did not show great differences in the evaluated parameters (Table 2)”  is not true. We are dealing with a very large difference in cholesterol content between CD - 18.10 and AF groups (13.33 and 11.65 mg / 100 g). It's hard to believe that it is not statistically significant !!!
• I think that in conlusions it should be mentioned that the addition Acacia farnesiana pods had a positive effect on the content of cholesterol in milk.

Author Response

Referee: 2

Comments to the Author

A very interesting scientific article worth publishing.

Critical remarks:

• Give how the chemical composition of milk was analyzed (Table 2)

Answer: the modification was made according to the reviewer´s suggestion.

• Why was there no fat concentration in the milk analyzed (Table 2)?

Answer: ether extract concept was eliminated. Now the fat concept is included in the table.

• How was the cholesterol content in milk determined?

Answer: Methodology of cholesterol determination is now mentioned in the material and method section.

• L211-L212: The sentence “The goats' milk composition did not show great differences in the evaluated parameters (Table 2)” is not true. We are dealing with a very large difference in cholesterol content between CD - 18.10 and AF groups (13.33 and 11.65 mg / 100 g). It's hard to believe that it is not statistically significant!!!

Answer: certainly true. In the previous version, this information was omitted. Now we present the differences among treatments and additionally a paragraph was added in the discussion and conclusion section dealing with cholesterol.

• I think that in conclusions it should be mentioned that the addition Acacia farnesiana pods had a positive effect on the content of cholesterol in milk.

Answer: the suggestion here is well taken. Now, the idea was included in the conclusion section.
